# Novel Fermentation Strategies of Strawberry Tree *Arbutus unedo* Fruits to Obtain High Nutritional Value Products

**DOI:** 10.3390/ijms25020684

**Published:** 2024-01-05

**Authors:** Francesca Anna Ramires, Miriana Durante, Isabella D’Antuono, Antonella Garbetta, Angelica Bruno, Annamaria Tarantini, Antonia Gallo, Angela Cardinali, Gianluca Bleve

**Affiliations:** 1Consiglio Nazionale delle Ricerche, Istituto di Scienze delle Produzioni Alimentari, Unità Operativa di Lecce, 73100 Lecce, Italy; francesca.ramires@ispa.cnr.it (F.A.R.); miriana.durante@ispa.cnr.it (M.D.); annamaria.tarantini@ispa.cnr.it (A.T.); antonia.gallo@ispa.cnr.it (A.G.); 2Consiglio Nazionale delle Ricerche, Istituto di Scienze delle Produzioni Alimentari, 70126 Bari, Italy; isabella.dantuono@ispa.cnr.it (I.D.); antonella.garbetta@ispa.cnr.it (A.G.); angelica.bruno@ispa.cnr.it (A.B.); 3Department of Soil, Plant and Food Sciences (Di.S.S.P.A), University of Bari, 70126 Bari, Italy

**Keywords:** strawberry tree, fermentation, yeast starter, functional food, antioxidant activity

## Abstract

The strawberry tree (*Arbustus unedo*) is a medicinal plant and an important source of biocompounds, potentially useful for pharmaceutical and chemical applications to prevent or treat several human diseases. The strawberry tree fruits have usually been used to produce traditional products such as jams and jellies and to obtain fermented alcoholic drinks, representing the most valuable derivative products. Other fermented products are potentially interesting for their nutritional value; however, the fermentation process needs to be controlled and standardized to obtain high-quality products/ingredients. In this work, we investigated two different fermentative procedures, using strawberry tree whole fruit and fruit paste as matrices inoculated with a selected starter strain of *Saccharomyces cerevisiae* LI 180-7. The physical, chemical, microbiological and nutritional properties of fermented products were evaluated, as well as their antioxidant activity. The new obtained fermented products are enriched in organic acids (acetic acid varied from 39.58 and 57.21 mg/g DW and lactic acid from 85.33 to 114.1 mg/g DW) and have better nutritional traits showing a higher amount of total polyphenols (phenolic acids, flavonoids and anthocyanins) that ranged from 1852 mg GAE/100 g DW to 2682 mg GAE/100 g DW. Also, the amount of isoprenoid increased ranging from 155.5 μg/g DW to 164.61 μg/g DW. In this regard, the most promising strategy seemed to be the fermentation of the fruit paste preparation; while the extract of fermented whole fruits showed the most powerful antioxidant activity. Finally, a preliminary attempt to produce a food prototype enriched in fermented strawberry tree fruits suggested the whole fruit fermented sample as the most promising from a preliminary sensory analysis.

## 1. Introduction

*Arbutus unedo* L., commonly known as the strawberry tree, is an economically important evergreen shrub of the Ericaceae family [1]. The *Arbutus* genus includes 12 species distributed across the west coast of the United States and Canada, Central America, western Europe, the Mediterranean Basin, North Africa, and the Middle East [2,3].

This shrub is ecologically relevant in southern European forests due to its resilience to abiotic and biotic stress factors [1].

Over the years, production areas of strawberry trees, previously considered a neglected species, have increased, particularly in Portugal, which is at present the world’s largest producer of these fruits [4,5].

The strawberry tree fruit is an edible spherical red berry, about 2 cm in diameter. The most abundant chemical components are represented by carbohydrates, with 40% of the total weight of fresh berries. In particular, sucrose is much more abundant in unripe fruits, while fructose is the main free sugar in the ripe phase [6]. Regarding the fatty acid composition, polyunsaturated fatty acids (PUFA) represent nearly 60% of the total, providing a high and favorable ratio of omega-3 to omega-6, due to the significant level of α-linolenic acid [7]. It has been reported that the fruits are characterized by the tasty flavor in their advanced maturation stage and the sensory characteristics derived from the combination of fatty acids and sugars. As a result, the berries are traditionally used to produce alcoholic beverages [8]. Furthermore, the fruit is an excellent source of vitamin C, dietary fiber, phenolic compounds (quinic acid, ferulic acid, protocatechuic acid, ellagic acid, gallic acid, caffeic acid, cinnamic acid, quercetin and myricetin), ascorbic acid and other fat-soluble antioxidants (lutein, α-tocopherol, β-carotene) [7,8,9].

The strawberry tree is a medicinal plant and a source of new molecules with high antioxidant potential. Polyphenols, commonly regarded as antioxidant compounds, play a major role in maintaining health [10].

This activity is owing to the high flavonoid content present in *A. unedo* (cyanidin, proanthocyanidins and delphinidin glycosides), vitamin C and E, as well as carotenoids, ellagic acid and its di-glucoside derivative [11].

Due to the presence of multiple bioactive compounds, strawberry tree fruits have been reported to exert therapeutic effects such as antiviral action against influenza [12,13] and HIV [14] and a reduction in cardiovascular disease risk [15]. Recent studies have also revealed that extracts of strawberry tree fruits inhibit the activity of α-amylase in the treatment of type 2 diabetes [6].

In addition, plant extracts have shown therapeutic potential against Alzheimer’s disease and Parkinson’s disease thanks to the presence of the glycoside arbutin, an inhibitor that blocks the progression of the disease [8]. Arbutin is commonly used in the cosmetic industry for skin whitening and is also found in commercial remedies for the treatment of urinary infections, often in combination with its precursor hydroquinone [16,17]. Thanks to their bioactivities, the natural compounds in strawberry tree fruit have been indicated as potentially useful for pharmaceutical and chemical applications to prevent or treat several human diseases [1].

Although the consumption of wild fruits has been shown to improve nutritional absorption and natural antioxidant levels [18], fruits are rarely consumed fresh, because of their high seed content, which is not appreciated by modern consumers, despite their pleasant flavor when fully ripe [19]. However, they have been used in folk medicine since ancient times for different purposes due to their healthful properties [20]. For this reason, the Food and Agriculture Organization (FAO) aims to increase the use of this precious wild species [21] to preserve plant biodiversity [22].

Owing to the high content of sugars, the fruits have been used in the production of traditional products such as jams and jellies and to obtain fermented alcoholic drinks, which represent the most valuable derivative products [1]. In the Algarve region (Portugal), the fully ripe fruits are fermented to produce a traditional alcoholic distillate called “Aguardente de medronho” (medronho firewater). The strawberry tree fruit fermentation is a natural process almost 36 days long, carried out by the associated yeast population, mainly *Saccharomyces cerevisiae* [23]. However, the results of fermentation, usually conducted in farms, are very variable in quality, with acidity issues, a lack of flavor or even the presence of off-flavors, because of uncontrolled fermentation (incomplete fermentation or overfermentation). Conversely, an interesting approach of solid-state fermentation driven by a selected yeast of *S. cerevisiae* was applied to berry pulp [24]. The fermented product was used for a subsequent distillation, obtaining a high-quality distilled alcoholic beverage. Studies conducted by Tejedor-Calvo and Morales [25] on newly formulated kombucha beverages also containing strawberry tree fruits treated with three different SCOBYs (symbiotic culture of aerobic mesophilic microorganisms, lactic acid bacteria, acetic acid bacteria and yeasts) for 21 days indicated that strawberry tree kombucha may be a flavored and nutritious drink, with the fermentation time and SCOBY composition identified as key factors to obtain the desired product. 

Furthermore, Santo et al. [26] studied the diversity of yeast populations during solid-state industrial fermentations of strawberry tree berries. One of the main results of this study was that starter cultures can be obtained by the microbiota and can be promoted to control the fermentation of strawberry tree fruits.

To the best of our knowledge, no studies have been conducted concerning the fermentation of strawberry tree fruit or paste directly to produce new ingredients/products. As already reported for other vegetable products, process control and standardization are necessary to improve fermentation and produce consistent, high-quality end products [27].

In this work, for the first time, we investigated two strategies for the fermentation of strawberry tree fruits, driven by a selected and well-characterized yeast strain for its specific biotechnological features, for producing novel fermented products. The main goal of this study was to investigate the use of microbial-driven fermentation as a tool to stabilize the raw material and to explore for the first time a new way for strawberry tree fruit valorization. Two different strawberry tree fruit preparations, the whole fruits (STWF) and a paste preparation obtained by grinding the fruits (STFP), were produced as matrices to be fermented. The fermentation process was monitored for 12 days in terms of physical, chemical, microbiological and nutritional properties. The total antioxidant activity of the obtained products was evaluated on intestinal cell lines. Food prototypes enriched with the two fermented products were produced and their associated aromatic profiles were preliminary elaborated. 

## 2. Results and Discussion

### 2.1. Strawberry Tree Fermentation Strategies

In this study, the yeast strain *S. cerevisiae* LI 180-7 [28,29,30] was tested as a starter candidate to ferment two different strawberry tree fruit preparations: the whole fruits (STWF) and a paste preparation obtained by grinding the fruits (STFP). 

The choice to use a *S. cerevisiae* strain in our study was based on the successful fermentation of strawberry tree fruit pulp in solid-state conditions, resulting in a distillate, as reported in the work of Gonzalez et al. [24]. Also, Cavaco et al. [23] studied the spontaneous fermentation of strawberry tree fruits to obtain a traditional spirit, discovering that the process was essentially made by yeast populations, with *S. cerevisiae* being the dominant species.

The fermentation processes were conducted for 12 days at room temperature (18–22 °C) in the dark to preserve the light-sensitive bioactive compounds. In both fermentations, the initial pH value was 3.61 and remained low, around 3.14–3.21, until the end of the process, (Table 1), in line with values reported for strawberry tree fruits harvested from different locations [24,31] and for a strawberry-tree-based kombucha fermented with symbiotic cultures of bacteria and yeasts [25].

The initial acidity of the fruits and pulp favored the adaptation and growth of the yeasts, as indicated by the total counts detected at Days 0, 4, 8 and 12 during fermentation, as reported in Table 2. The ability of the inoculated starter to grow and drive the fermentation process in both the STWF and STFP samples was confirmed by the increase in the yeast total count after 4 days of inoculation with continuous increments until the end of the process. These data agree with those reported by Cavaco et al. [23] on the spontaneous fermentation process, mainly due to the yeast population.

The analysis in Table 2 also revealed a low presence of molds in both samples only in the initial stage, likely limited by the fermentative process afterwards. An aerobic colony count showed a different behavior between the two fermentation strategies. In the STWF sample, the count increased throughout the process, reaching a plateau after 4 days of fermentation. In contrast, in the STFP sample, there was a rapid decrease in this microbial population, thus suggesting a strong ability of the yeasts to colonize the biological niche and use almost all of the available sugars. As expected, in the STFP sample, lactic acid bacteria (LAB) were not detected during the entire process, probably due to a different environment produced by the fermentation of the fruit paste. Interestingly, in the STWF sample, lactic acid bacteria (LAB) appeared after 4 days of fermentation at very high count levels (>7 Log_10_ CFU/g), thus suggesting that the yeasts favored, with their presence and metabolism, the development of autochthonous LAB. 

The pH value remaining below 4 in both fermented samples was adequate to ensure the absence of microbial contaminants. In the samples analyzed, no potential pathogens or spoilage microorganisms were detected, including *Clostridium* spp., Enterobacteriaceae, *E. coli*, *Aerogenes* bacteria and putative pathogenic *Staphylococcus* spp. (Table 2). 

The correct trend of the fermentation process was also confirmed by the reduction in sugar content in both samples during fermentation, as recorded on Days 0, 4, 8 and 12 (Brix % value in Table 1). 

The initial glucose + fructose content (82.44 ± 0.32 mg/g for STWF and 199.1 ± 4.6 mg/g for STFP) decreased by up to 22% in the STWF sample and was completely consumed in STFP after 12 days (Figure 1A).

A specific profile of organic acids was obtained for each sample, significantly influenced by the preparation of the raw material. In the STWF strategy, the organic acids generally increased during the fermentation process, reaching the maximum concentration after 12 days, in concordance with the decrease in sugars (Figure 1B–D). The acetic acid and the lactic acid increased from 28.48 ± 0.16 to 57.21 ± 0.78 mg/g DW and from 38.50 ± 5.22 to 85.33 ± 10.49 mg/g DW, respectively. The citric acid increased until Day 8 and then decreased to the initial value. 

Generally, the STFP sample showed an initial content of organic acids higher than the STWF sample, likely due to their release from ground vegetable material, with a lower increase observed during the fermentation. In STFP, the maximum concentrations of acetic and lactic acid were obtained after 8 days of fermentation (39.58 ± 0.15 mg/g DW and 114.1 ± 2.26 mg/g DW, respectively), while citric acid reached its maximum level after 4 days of fermentation (79.60 ± 4.25 mg/g DW). This last evidence differed from previous studies that reported the absence of quantifiable amounts of citric acid in strawberry tree fruits [31,32].

It can be assumed that the initial concentrations of organic acids observed in our study depend on the plant material. In contrast, the increase in lactic and acetic acids probably arises from microbial activity [28], as well as yeasts and heterofermentative LAB which are able to generate acetic acid from fermentable material under environmental stress and metabolize citric acid [33].

The results obtained in the current work provide for the first time qualitative and quantitative information on the presence of organic acids in fermented strawberry tree fruit since the data reported thus far have regarded volatile compounds in arbutus berry distillates [24] and in strawberry tree kombuchas fermented with symbiotic cultures of bacteria and yeasts [25]. 

### 2.2. Phenolic Compound Content

The samples of strawberry trees have been analyzed for the assessment of the total phenols content (TPC), total flavonoids content (TFC) and total anthocyanins content (TAC). Figure 2A shows the data related to TPC highlighting that the fermentative process positively influenced both sample forms (whole and paste) and the recovery of polyphenols. In particular, the fermented fruit paste showed the higher amount (2682 mg GAE/100 g DW) followed by fermented whole fruits (2406 mgGAE/100 g DW), fruit paste (2185 mgGAE/100 g DW) and whole fruits (1852 mgGAE/100 g DW). This behavior was also confirmed for the other two classes of polyphenols, flavonoids (from 286 mg CAT/100 g DW to 407 mg CAT/100 g DW) and anthocyanins (from 12 mg C3G/100 g DW to 15 mg C3G/100 g DW), respectively, as reported in the Figure 2B,C, while small differences were recorded between the two forms of non-fermented samples. Although the phyto-chemical composition of strawberry tree fruits could be influenced by the ripening stage, the cultivars and the environmental and climatic conditions [34], the results reported in this study are in good agreement with those published by other authors. In particular, Zitouni, et al. [10], for some *Arbutus unedo* L. cultivars characteristic of Morocco, reported average total polyphenols of about 32.22 mg GAE/g DW, total flavonoids of about 5.20 mg RE/g DW and total anthocyanins of about 0.40 mg cya-3-glu/100 g DW, confirming that the strawberry tree is a rich source of phenolic compounds with a wide range of health-promoting effects and suitable application as a functional ingredient [35].

To obtain more information related to the polyphenol compounds present in the hydro-alcoholic extracts of ST samples, an HPLC-DAD analysis was performed and the results are shown in Table 3.

From the analysis. 16 compounds have been identified with gallic acid and its derivatives, ellagic acid and its derivatives and cyanidin 3-O-glucoside as the main compounds, followed by two glycosides in the form of quercetin and quercetin aglycone. The HPLC-DAD profiles of strawberry tree fruits showed a similar phenolic composition as reported in previous studies [10,11] with some differences probably related to the different cultivars and stages of ripening used. The sample preparations (whole and paste) and the biological treatments (fermentation and non-fermentation) significantly influenced the phenolic composition. Therefore, the presence of the main identified polyphenols was higher in the whole sample compared to the paste one, in which some minor compounds (myricetin and kaempferide) were undetectable. The fermentation process, instead, improved the polyphenols extractability in the paste sample by about two times compared to the non-fermented control. The highest increase was attributable to gallic acid, ellagic acid and quercetin aglycone, although a slight decrease in anthocyanins was also recorded. Furthermore, in the whole sample, the fermentation slightly reduced the polyphenol content compared to the unfermented control. Further, as reported in fermented polyphenol rich-foods, the combination with the processing influences the polyphenol amount, improving the bioavailability and boosting the growth of the beneficial bacteria inhibiting the pathogens [36].

### 2.3. Isoprenoids Content during Fermentation of Strawberry Tree Fruit Samples

The STWF and STFP samples were also assayed for the variation in the composition of isoprenoids (tocopherols and carotenoids) during the fermentation process at Days 0. 4, 8 and 12, as depicted in Figure 3. Isoprenoids include a group of natural products that show beneficial biological activities, finding applications in the pharmaceutical, nutraceutical and flavor fragrance industries, among others. Strawberry tree fruits are a source of α-tocopherols, the most biologically active form of vitamin E, followed by β-carotene (precursor of vitamin A), lutein and zeaxanthin.

During the fermentation of strawberry tree berries, α-tocopherol content in whole fruit (123.1 ± 2.6 µg/g DW) and fruit paste (84.7 ± 5.9 µg/g DW) increased by 1.3- and 1.8-fold, respectively, after 4 days of fermentation (164.61 ± 6.1 µg/g DW and 155.5 ± 7.5 µg/g DW in whole fruit and fruit paste, respectively), and it did not present statistically significant differences until the end of the process (Figure 3A,B). A similar trend was observed for β-carotene in paste fruit (Figure 3B). In the whole fruit preparation, the β-carotene content increased during the first 4 days of fermentation, followed by a slight decrease after 8 days (Figure 3A). The xanthophylls, such as lutein and zeaxanthin, present in low amounts, showed an increase in both samples after 4 days of fermentation compared to the initial value. In the paste fruit sample, xanthophylls showed the highest content at the end of fermentation, while in the whole fruit, lutein and zeaxanthin presented the highest values after 8 and 4 days of fermentation, respectively. In general, it is supposed that the increase in α-tocopherol and carotenoids in fermented strawberry tree berries may be related to structural changes induced by fermentation, which may increase the ability to extract isoprenoids [37].

### 2.4. Enzyme Activities

The enzyme activities identified in fermented vegetable matrices include both those originating in vegetable tissues and those produced by the microbial component during the fermentation process [38,39]. Moreover, the enzyme stability and concentrations change during fermentation depending on various factors including temperature, pH, microbial consortia evolution and the production of potential inhibitors.

Since the enzymatic activities can contribute to the flavor, texture, safety and nutritional traits of the final products, in this study, six of the main enzyme activities were tested on extracts obtained from the STWF and STFP starting material (T0) and the fermented samples at the end of the process (T12). The yeast-driven fermentation produced a statistically significant improvement of lipase and esterase activities in the STWF sample, with a corresponding reduction in xylanase activity (Table 4). On the other hand, the esterase and the xylanase activities increased by about 1.7 times after fermentation in the STFP sample compared with the corresponding starting raw material. Esterase and lipase activities can improve the taste of fermented products by originating esters from free fatty acids and changing their composition in the final product [40].

Among xylanases, endo-xylanases and β-xylosidases are the two most relevant enzymes responsible for the hydrolysis of the main component of hemicellulose, xylan [41,42].

To investigate the possible effect of hydrolyzing enzyme activities tested in the fermented products on the content of several bioactive compound classes, a correlation matrix analysis was completed in Table 5 using the T0 and T12 data obtained for each treatment. The total phenol, flavonoids and anthocyanin content revealed a positive correlation with esterase and protease. Indeed, the proteases produced during fermentation were effectively involved in the mobilization of phenolic compounds [43].

Earlier studies have demonstrated that most phenolic compounds in vegetable tissues are covalently bound to cell wall structural components (cellulose, hemicellulose, lignin, pectin, structural proteins, etc.) and are difficult to extract [44,45].

The total anthocyanin content was also positively correlated with cellulase activities, as already demonstrated in other studies where cellulases were able to hydrolyze insoluble-bound phenolics linked with structural protein/carbohydrates via ester linkages and with lignin via ether linkages [45,46]. A positive correlation was also observed between isoprenoid content and amylase activities. This last evidence suggests the enzymatic liberation of soluble phenolics could be an energy-demanding process, with the exception of beta-carotene, which also showed a good correlation with lipase [44,47].

### 2.5. Antioxidant Activity

When tested on the Caco-2 intestinal cell line, the samples STWF and STFP, before and after 12 days of fermentation (T0 and T12), showed a high and dose-dependent antioxidant activity without any detectable differences, as reported in the CAA unit values in Table 6.

Strawberry tree fruit, especially due to its antioxidants, can be considered a “health-promoting food”. Antioxidants are highly requested for their health benefits provided by natural compounds preventing the occurrence of oxidative-stress related diseases, caused by free radicals attacking lipids or nucleic acids.

However, the in vitro biological activity, especially the antioxidant activity, has to be tested [48]. Tenuta et al. [49] have tested different *A. unedo* extracts (different extraction techniques) for their potential free radicals scavenging activity by ABTS (2,2-azinobis (3-ethylbenzothiazoline-6-sulfonic acid) and DPPH (2,2-diphenyl-1-picrylhydrazyl) assays. The antioxidant properties of the extracts were concentration-dependent, in agreement with our findings in this study.

In order to compare the antioxidant activity of different extracts, the CAA assay involves the determination of the median effective dose (the concentration of bioactive compounds that induce a 50% effect), with a lower median effective dose corresponding to a higher antioxidant activity.

Analyzing the median effective doses in Figure 4, it appears that for STWF, after fermentation (T12), the median effective dose was lower than T0, and the fermented extract exhibits a strong antioxidant activity (1.22 ± 0.14 vs. 0.85 ± 0.105 µg total phenolic fraction (TPF)/g DW), although the difference is not statistically different. On the contrary, the STFP at T0 appears to have the highest antioxidant activity, with a median effective dose of 0.40 ± 0.075 µg TPF/g DW.

On the contrary, for STFP after 12 days of fermentation, the median effective dose was five times higher (*p* < 0.05) than T0 (0.40 ± 0.075 vs. 2.12 ± 0.52 µg TPF/g DW), and this extract appears to be the less potent extract despite the highest concentration of total polyphenols. It can be supposed that the differences in the matrix structures and formulation, due to the different preparations, influenced the fermentative process and the composition of the bioactive compounds produced.

### 2.6. Principal Component Analysis

An overview of the correlation between the main biochemical and physical parameters, enzyme activities and the phenolic changes and isoprenoid content was investigated using a principal component analysis (PCA) model, represented in Figure 5. The result showed that the first two validated principal components (PCs) attributed to 83.26% of the total information in terms of the original variables, could well describe the information of the original variables.

The samples treated with the inoculated yeast strain (T12) were evidently separated from the untreated ones (T0). In addition, the STFP samples were completely separated from the STWF ones in the plane. The STFP fermented (T12) sample clustered with TPC, protease, lipase and esterase activities and with carotenoids, as well as with lactic and citric acids, whereas the corresponding sample T0 grouped with TAC, TFC, cellulase and glucose + fructose content. The STWF fermented sample clustered in the plane associated with amylase, xylanase and vitamins. Finally, the STWF T0 sample was located in the opposite portion of the plane, associated with pH, as a unique variable.

The most promising strawberry tree fruit strategy to be used in the fermentation process seemed to be the STFP, depending on the developed desired end-products, although from the antioxidant activity analysis, it appeared to be the less powerful potent extract despite the highest concentration of total polyphenols.

### 2.7. Preparation and Descriptive Sensory Analysis of an Enriched Food Prototype

In this study, a preliminary test for the potential positive role of STWF and STFP fermented products as new ingredients for introducing new sensorial traits in food products was attempted. To this end, biscuits containing approximately 18% (*w*/*w* FW) were prepared. The obtained products were submitted to a sensory analysis. Since no established descriptors have been already established for the sensory attributes of these fermented products, the most promising acceptable flavor traits were preliminarily established by voluntary panelists with a preliminary aroma sensory evaluation of the biscuits.

This analysis revealed that the obtained products had significantly different flavor profiles (Figure 6).

In particular, the biscuits added with the fermented STWF sample were perceived with the highest scores for all the tested notes, that is, nuts, berries, chocolate, vanilla, cinnamon, must and exotic fruit, except for the sandalwood note slightly prominent in the biscuits added with fermented STFP. Finally, the apple and citrus notes were identified at the same level in both biscuits.

The use of yeasts can be suitable to generate characteristic flavor compounds in strawberry tree fruits-derived products, especially due to the alcoholic fermentation and their ester biosynthesis pathways [24].

## 3. Materials and Methods

### 3.1. Fruit Sampling

The strawberry tree fruits (*Arbutus unedo* L.) used in this study were obtained from trees grown at the University of Salento (Lecce, Italy), in November and December 2021. Fruits were harvested at the intermediate/final ripening levels (14–16 °Brix). Fresh fruits were immediately transported to the laboratory for the subsequent preparatory processes for fermentation.

### 3.2. Microbial Strain

In this study, the yeast strain *S. cerevisiae* LI 180-7 (DSM 27800), previously isolated from fermented black table olives, was chosen for its demonstrated capacity to drive the fermentation of vegetable matrices, also under hard conditions, such as high salt and phenol concentrations [28,29,30].

### 3.3. Preparation of Samples and Fermentation Conditions

The fruits were extensively washed with drinking water. After the removal of water, 2 different preparations of strawberry tree fruits were submitted to lab-scale fermentation according to the following procedures:

Strawberry tree whole fruits (STWF): Approximately 0.5 kg of whole fruits was placed in a 1 kg-capacity glass jar and drinking water was added in a ratio of 1:1 (*v*/*v*). Subsequently, the starter strain *S. cerevisiae* LI 180-7 was added at a concentration of about 10^7^ CFU/g final volume (1 kg).

Strawberry tree fruit paste (STFP): Approximately 1 kg of fruit was ground using a blender and placed in a 1 kg-capacity glass jar. Subsequently, the previously prepared starter strain *S. cerevisiae* LI 180-7 [28] grown in Sabouraud liquid medium was added at a concentration of about 10^7^ CFU/g final volume (1 kg).

The fermentation process was carried out at room temperature (18–22 °C) in the dark to preserve the light sensitive bioactive compounds for 12 days. Approximately 15 g of samples was collected at different time points (0, 4, 8 and 12 days) and stored for the subsequent analyses.

### 3.4. Physical, Chemical and Microbiological Analyses

The refractive index (Brix) and pH were evaluated at Days 0, 4, 8 and 12 of fermentation. Brix concentration was measured by using a Brix refractometer for food at 0–18% Brix with automatic temperature compensation: RHS-MR110 ATC (Giorgio Bormac srl, Carpi, Mo, Italy).

Glucose and fructose, lactic acid and acetic acid in whole fruits and paste were analyzed using a Miura enzymatic analyzer (Exacta Optech, Modena, Italy), as already reported by Guzzon et al. [50] and Catalano et al. [51].

For the microbiological assays, 1 g of each sample, obtained as described previously at different time intervals, was mixed with 9 mL of sterile peptone water and decimal solutions were prepared. The total bacterial count was carried out using plate count agar (PCA, Heywood, Lancashire, UK) added with 0.05 g/L nystatin (Sigma-Aldrich, Darmstadt, Germany) and incubated at 30 °C for 48–72 h; coli–aerogenes bacteria detection and enumeration by violet red bile agar (VRBA, LABM, Heywood, Lancashire, UK) incubated at 37 °C for 24–48 h; sulfite-reducing clostridia by sulfite polymyxin sulphadiazine agar (S.P.S. Sigma-Aldrich, Darmstadt, Germany); the enumeration of coagulase-positive staphylococci by Baird Parker agar base (BP, LABM, Heywood, Lancashire, UK) incubated at 37 °C for 24–48 h; and Enterobacteriaceae identification was performed by violet red bile glucose agar (VRBGA, LABM, Heywood, Lancashire, UK) incubated at 37 °C for 18–24 h. For the enumeration of lactic acid bacteria (LAB), de Man, Rogosa and Sharpe agar was used, added with 0.05 g/L nystatin (Sigma-Aldrich, Darmstadt, Germany) and incubated at 30 °C for 48–72 h. Yeast and mold total counts were executed on dichloran Rose–Bengal chloramphenicol agar (DRBC, Thermo Fisher Scientific, Monza, Italy) and incubated at 25 °C for 5 days.

### 3.5. Extraction and Analysis of Phenolic Compounds

Samples were freeze-dried until reaching a stable weight (about 40 h) using an Alpha 2-4 LSC plus freeze-dryer (Martin Christ Gefriertrocknungsanlagen GmbH, Osterode am Harz, Germany) with a vacuum pressure of 0.015 mbar and a condenser temperature of −80 °C. The freeze-dried fruits were ground at 500 µm using a Retsch laboratory mill (Torre Boldone, Bg, Italy) to obtain a powder.

Lyophilized strawberry tree fruit samples, STWF and STFP, non-fermented (T0) and after 12 days of fermentation (T12), were extracted with 80% acidified methanol (methanol/water/trifluoroacetic acid 80:19.5:0.5) by ultrasonic water bath (Fisherbrand™ P-Series, Fisher Scientific, Monza, Italy) for 30 min, frequency 37 kHz), at room temperature and at a power setting of 50%. The extraction was performed twice, adding fresh solvent to the pellets after centrifugation, and the supernatants were collected, measured as final volume and utilized for further analysis.

The determination of total phenolic content (TPC) in the extracts was performed by Folin–Ciocalteu (FC) assay as described in Ramires et al. [52], and TPC was expressed as mg gallic acid equivalent (GAE), per 100 g dry weight of strawberry tree samples. The total flavonoid content (TFC) was determined according to Jia et al. [53]. Catechin (CAT) was used as a standard and the results were expressed as mg of CAT equivalent per 100 g of strawberry tree fruit dry weight.

The total anthocyanins concentration (TAC) was determined by a pH-differential method following the procedure reported by Giusti and Wrolstad [54]. The dilution factor for the extracts was previously determined and was 1:2. The TAC, in the original samples, was quantified in mg of cyanidin 3-O-glucoside (C3G) equivalent in 100 g of DW, using the following equation:$$Monomeric\;anthocyanin\;pigment=\frac{A\times MW\times DF\times 1000}{\left(\varepsilon \times 1\right)}$$
where *A*: Absorbance = (A510 nm–A700 nm) pH 1.0—(A510 nm–A700 nm) pH 4.5; *MW*: molecular weight (449.2 g/mol); *DF*: dilution factor; *Ɛ*: molar absorptivity coefficient of cyanidin 3-O-glucoside (26,900 L/mol·cm).

The polyphenol profile was obtained by analyzing the strawberry tree extracts by HPLC-DAD, using the Agilent 1260 Infinity Series Chromatograph system (Palo Alto, CA, USA). The instrument was equipped with a 1260 HIP Degasser, G1312B Binary Pump, G1316A Thermostat, G4212B DAD Detector and supplied with Agilent Open Lab CDS Chem Station Software. For separation, an analytical Luna C18 (4.6 × 250 mm; 5 μm) column (Phenomenex Torrance, CA, USA) was used. The analytical method and elution profile were described in D’Antuono et al. [55]. The identification was performed by comparing the spectra and retention time of the pure available standards; the phenolic compounds derivatives (gallic acid and ellagic acids, anthocyanins) were identified by UV–Vis spectra and quantified by the response factors of their pure standard reference.

### 3.6. Extraction and Analysis of Isoprenoids (Tocopherols and Carotenoids)

The isoprenoid content was evaluated in STWF and STFP at 0, 4, 8 and 12 days, during fermentation.

The isoprenoid determination was carried out according to Blando et al. [56]. Briefly, 50 mg of freeze-dried sample was extracted with a 3 mL mixture of hexane/ethanol/acetone (2/1/1 *v*/*v*/*v*) containing 0.05% of BHT. Samples were shaken on an orbital shaker at 180 rpm for 15 min. Then, 3 mL of distilled water was added and the suspension was centrifuged at 4500× *g* for 10 min. The organic phase was collected and dried under nitrogen, resuspended in 100 µL of ethyl acetate and analyzed using an Agilent 1100 Series HPLC system as described by Durante et al. [57].

### 3.7. Evaluation of Enzyme Activities

The activity of six enzymes (α-amylase, protease, esterase, lipase, cellulase and endo-xylanase) was assayed in STWF and STFP, non-fermented (T0) and at the end of fermentation (T12). All experiments were conducted in triplicate.

STWF and STFP crude enzyme solutions were prepared according to the method of Lee et al. [58] with slight modifications. Briefly, 2.5 g of each sample was suspended in 5 mL of distilled water and incubated with shaking at 1000 rpm for 1 h at 30 °C. Then, solid and liquid portions of the mixture were separated by filtering with a polyamide filter 355/51 (Saati, Milan, Italy). Next, the mixture was centrifuged at 9000 rpm at 4 °C for 15 min. Subsequently, the supernatant was removed to obtain the crude enzyme solution. Before further analysis, the resulting crude enzymes were dissolved in the appropriate buffers.

Lipase, esterase and endo-xylanase activity assays were carried out according to the procedure described by Maiorano et al. [38]; α-amylase and protease activity tests followed the method described by Ramires et al. [52]. Finally, cellulases were determined using a cellulase assay kit (CellG5, Megazyme, Bray, Ireland), according to the method provided by the manufacturer.

### 3.8. Antioxidant Activity

The cellular antioxidant activity of STWF and STFP, non-fermented (T0) and after 12 days of fermentation (T12), was determined using the Caco-2 intestinal cell line purchased from the European Collection of Authenticated Cell Cultures (ECACC). The Caco-2 cell line was grown in 25 cm^2^ flasks at a starting density of 250,000 cells/mL in Dulbecco’s Modified Eagle’s Medium with 4.5 g/L glucose supplemented with 10% of FBS, 1% of L-Glutamine, 1% antibiotic and antimycotic solution and 1% non-essential amino acid solution. Density and cell viability were determined by a Scepter automatic cell counter (Merck Millipore, Milan, Italy). The cells used for experimental protocols showed a mean viability of 90%.

The antioxidant activity of hydroalcoholic extracts (mix) of samples was measured as a reduction in intracellular induced reacting oxygen species (ROS) by applying the cellular antioxidant activity (CAA) assay. The CAA assay was performed according to the procedure described by Wolfe and Liu [59] with some modifications.

Briefly, the cellular suspension was seeded on a 96-well white flat-bottom plate and incubated at 37 °C for 24 h. After seeding, cells were stained with 5 μM of 2–7-dichloro-dihydrofluorescein diacetate (DCFH-DA) and incubated for 30 min. After the staining phase, cells were treated for 30 min with the hydroalcoholic extracts in the following concentration ranges: STWF T0, from 0.38 to 240 µg TPF/gr DW; STWFT12, from 0.36 to 224 µg TPF/gr DW; STPF T0, from 0.25 to 155 µg TPF/gr of DW; STPF T12, from 0.71 to 441 µg TPF/gr of DW.

Then, cumene hydroperoxide (12.5 μM) was added to the cells as a stress inducer, and the fluorescence (Ex 485 nm, Em 530 nm) was measured every 5 min for 1 h at 37 °C using a Varioskan Flash Spectral Scanning Multimode Reader (Thermo Fisher Scientific, Milan, Italy). This procedure allowed the mathematical calculation of different parameters, including the CAA unit and the median effective dose (EC50) as described by Garbetta et al. [60].

The CAA unit was measured by integrating the area under the kinetic curve (fluorescence vs. time). Higher values of CAA units indicate a high antioxidant activity. The EC_50_ corresponds to the concentration of polyphenols (µg TPF/gr DW) that produces a 50% reduction in induced ROS.

### 3.9. Preparation and Descriptive Sensory Analysis of the Food Prototypes Enriched in Fermented Strawberry Fruit Fermented Products

Biscuits were prepared as follows: 80 g sunflower oil, 100 g sucrose, one egg, 280 g type 00 wheat flour, 100 mL of water or 100 g fermented STWF or STFP (corresponding to 18% *w*/*w* DW). The obtained biscuits were baked for 20 min at 150 °C.

For the preliminary characterization of the sensory properties of the biscuits enriched in fermented strawberry fruit products, a sensory panel was made up of ten women and ten men (ranging from 30 to 70 years old). The biscuit samples were administered to the panelists in two sessions to select the best descriptors for several aroma attributes, and a third session was conducted to identify the intensity of the selected attributes/descriptors on a seven-point intensity scale (0—none; 1—delicate; 2—delicate to moderate; 3—moderate; 4—moderate to intense; 5—intense; 6—very intense). The results were the mean values of the two sensory sessions.

### 3.10. Statistical Analysis

Statistical analyses were performed using SigmaPlotTM software v.12 (Systat Software, Inc., SigmaPlot for Windows, San Jose, CA, USA). The all-pairwise multiple comparisons Dunn’s method was used to evaluate significant differences between cells treated with SF extracts. A one-way ANOVA test followed by the Tukey post hoc method was applied to establish significant differences between means (*p* < 0.05) in isoprenoid content. Data were expressed as the mean ± standard deviation of values from 3 independent experiments. Values of *p* < 0.05 were considered statistically different.

To establish significant differences among glucose and fructose, lactic acid, acetic acid and citric acid values, Kruskal–Wallis tests followed by Dunn’s multiple comparisons test (*p* < 0.05) were carried out, whereas enzyme activities were compared by the Mann–Whitney test using GraphPad Prism 6.0 software (La Jolla, CA, USA).

The mean values related to phenolic contents (TPC, TPF, TPA and HPLC analysis) were subjected to one-way ANOVA followed by LSD’s post hoc test. Results analysis was performed using the software STATISTICA 6.0 (StatSoft, Tulsa, OK, USA) and was considered significant for *p* ≤ 0.05.

A two-tailed correlation test with Spearman’s rank correlation coefficient (R) was calculated among the enzyme activities and main bioactive compounds.

Principal component analysis (PCA) was used to compare the microbiological, chemical and biochemical parameters associated with the fermented and unfermented samples using the OriginPro 2016 software (OriginLab, Northampton, MA, USA).

## 4. Conclusions

In conclusion, the yeast strain *S. cerevisiae* LI 180-7 was able to ferment both the different strawberry tree fruit preparations: the whole fruits (STWF) and a paste preparation obtained by grinding the fruits (STFP). The yeast-driven fermentation produced new fermented products enriched in organic acids and stabilized at the chemical and microbiological levels.

Up to now, *S. cerevisiae* was the first yeast used for strawberry tree fruit fermentation; however, the evidence reported in this study can pave the way for future investigations of the potential offered by other microbial species/strains or microbial consortia, autochthonous or allochthonous, with specific biotechnological and enzymatic features.

An enrichment in nutritional traits, such as total phenolic, flavonoid, anthocyanin and isoprenoid content, was observed after fermentation. The vitamin A and vitamin E contents in the STWF and STFP fermented products were increased in comparison with the raw samples, contributing to an improved vitamin status.

Specific correlations were reported among some enzyme activities and compounds involved in conferring potential functional traits to the products. Considering the developed desired end products, the most promising strawberry tree fruit fermentation strategy seemed to be the STFP, whereas the more powerful extract in terms of cellular antioxidant activity was the fermented STWF extract.

The improved content of total phenolic, flavonoid, anthocyanin and isoprenoid and the deriving antioxidant activity in fermented strawberry tree products can be highly promising for the future use of these ingredients for formulating enriched functional foods. In this direction, for the first time, the preparation of biscuits as a food prototype enriched in ST fermented products was attempted, revealing that the STWF sample achieved the highest scores for all of the tested notes (nuts, berries, chocolate, vanilla, cinnamon, must and exotic fruit).

## Figures and Tables

**Figure 1 ijms-25-00684-f001:**
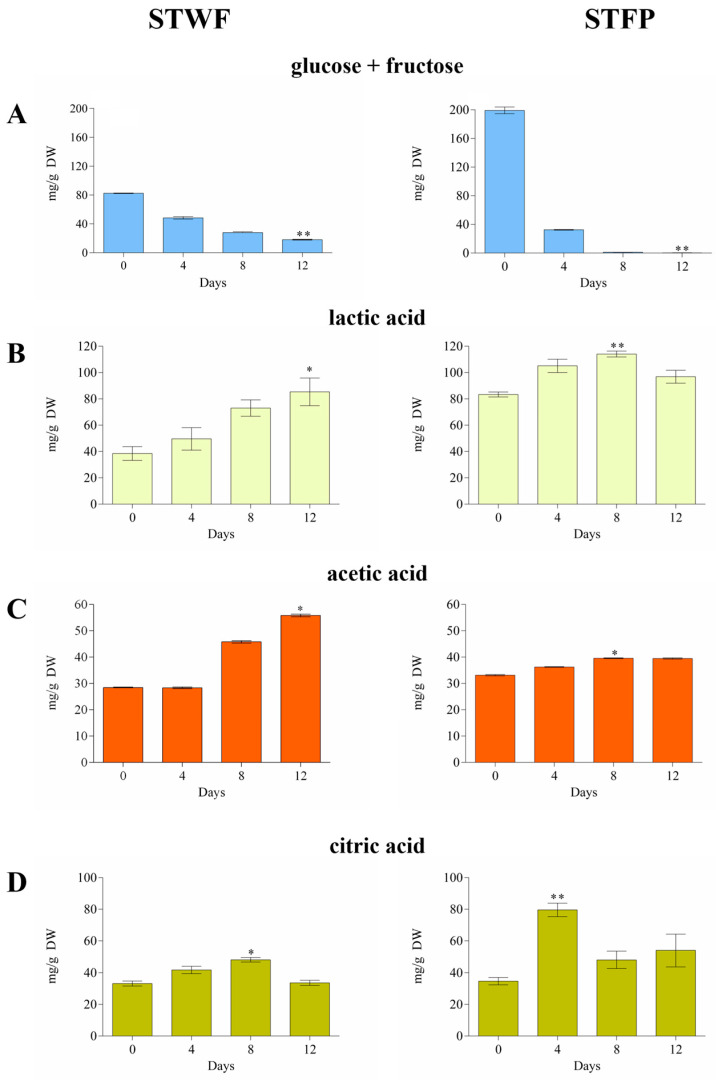
Consumption of glucose + fructose (**A**) and the evolution of lactic acid (**B**), acetic acid (**C**) and citric acid (**D**) during fermentation of the STWF and STFP samples. Values are the means of three independent measurements ± standard deviation. Kruskal–Wallis statistical tests followed by Dunn’s multiple comparison post hoc test were used to compare each treatment with the control (* *p* < 0.05 and ** *p* < 0.001).

**Figure 2 ijms-25-00684-f002:**
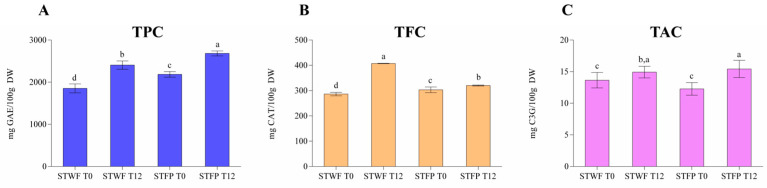
Total phenolic content (TPC) (**A**), total flavonoid content (TFC) (**B**) and total anthocyanin content (TAC) (**C**) in strawberry tree whole fruit (STWF) and strawberry tree fruit paste (STFP) not fermented (T0) and after 12 days of fermentation (T12). Letters indicate statistical differences according to the Kruskal–Wallis test followed by Dunn’s multiple comparison post hoc test. a–d: the different letters in lines indicate significant differences among samples during the fermentation process (*p* < 0.05).

**Figure 3 ijms-25-00684-f003:**
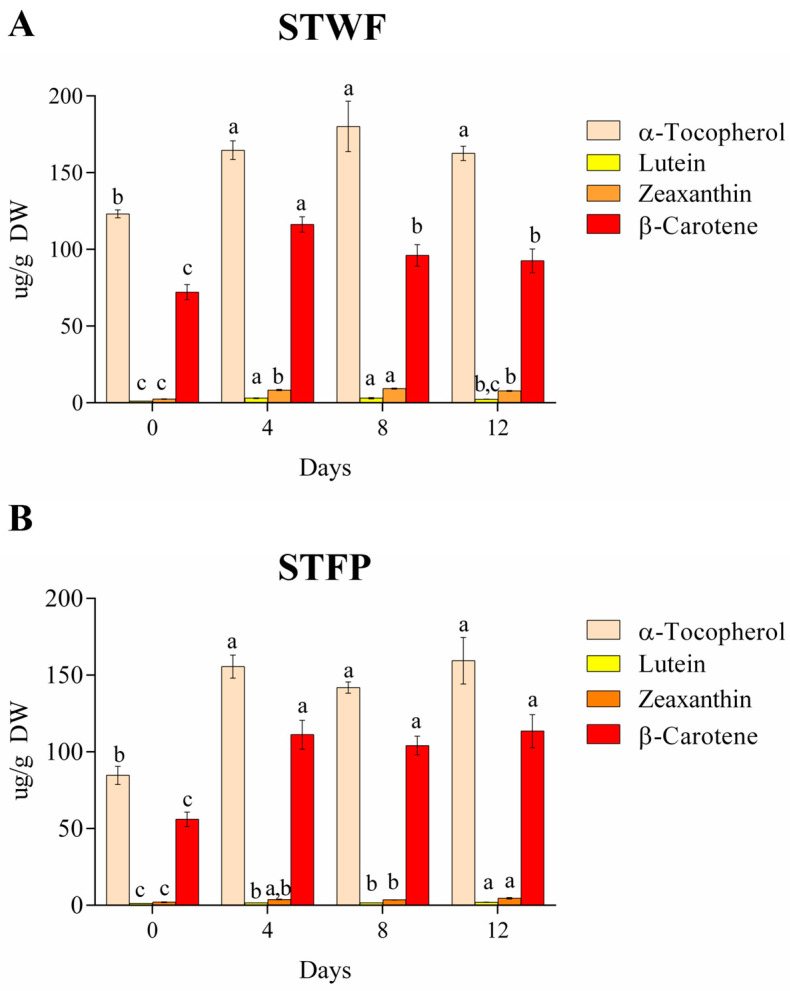
Evolution of isoprenoids on different days during fermentation of the STWF (**A**) and STFP (**B**) samples. Letters indicate statistical differences according to the Kruskal–Wallis test followed by Dunn’s multiple comparison post hoc test. a–c: the different letters in lines indicate significant differences among samples during the fermentation process (*p* < 0.05).

**Figure 4 ijms-25-00684-f004:**
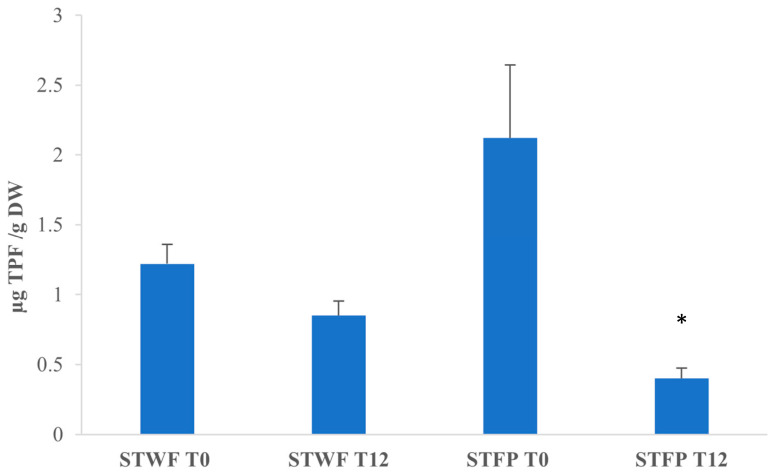
Median effective dose of strawberry tree fruits-derived samples. STWF: strawberry tree whole fruits; STFP: strawberry tree fruit paste. TPF: total phenolic fraction. Data are expressed as the mean ± SD of three independent experiments. *: *p* < 0.05 evaluated by all pairwise multiple comparisons using Dunn’s method between STWF T0 and T12 and between STFP T0 and T12.

**Figure 5 ijms-25-00684-f005:**
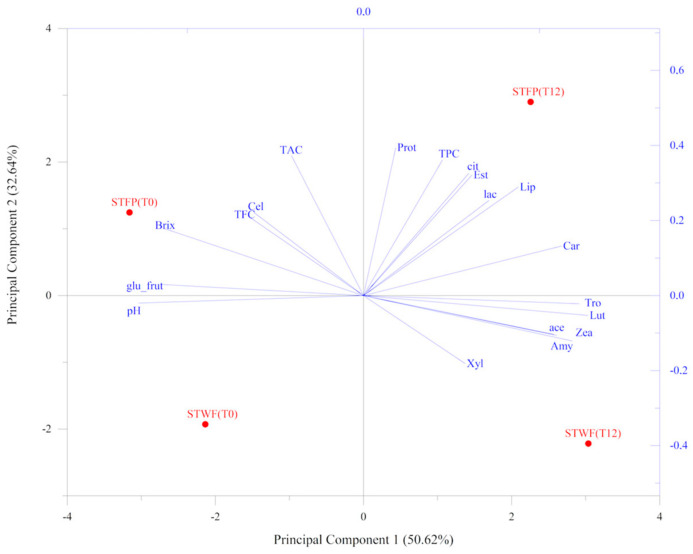
PCA analysis performed on parameters associated with strawberry tree fruit preparations. PCA variables were obtained from the analysis of nutritional traits, enzyme-associated activities and chemical composition values. STWF: strawberry tree whole fruits; STFP: strawberry tree fruit paste. T0: unfermented product; T12: fermented product for 12 days.

**Figure 6 ijms-25-00684-f006:**
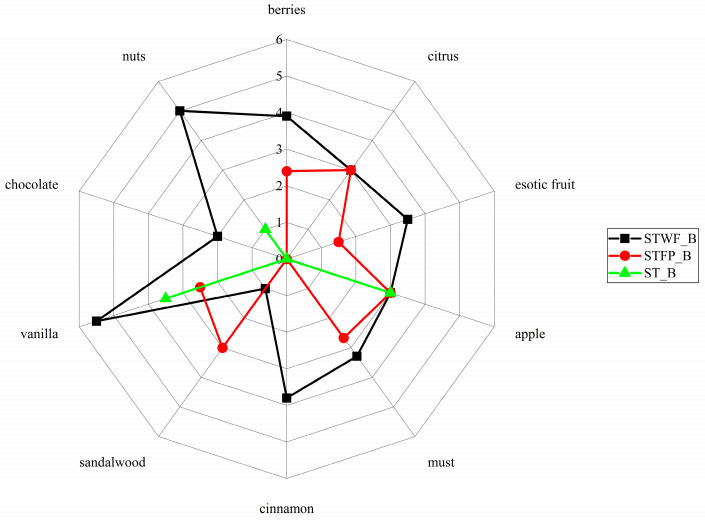
Effects of STWF and STFP fermented by a yeast starter culture on the aroma descriptors of biscuits. ST_B = Standard_Biscuit.

**Table 1 ijms-25-00684-t001:** Physico-chemical parameters associated to yeast driven fermentation of strawberry tree whole fruits (STWF) and strawberry tree fruits paste (STFP) at 0, 4, 8 and 12 days.

Time of Fermentation (Days)	STWF		STFP	
	pH	Brix (%)	pH	Brix (%)
0	3.61 ± 0.51 a	10.5 ± 0.6 a	3.61 ± 0.57 a	15.9 ± 0.6 a
4	3.45 ± 0.40 a	8 ± 0.5 b	3.49 ± 0.61 a	12 ± 0.6 b
8	3.23 ± 0.43 a	6.8 ± 0.3 b,c	3.28 ± 0.32 a	11 ± 0.4 b
12	3.14 ± 0.49 a	5.2 ± 0.3 c	3.21 ± 0.44 a	9.2 ± 0.4 c

Values are the means of three independent measurements ± SD. Letters indicate statistical deviation according to the Kruskal–Wallis test followed by Dunn’s multiple comparison post hoc test. The different letters in columns indicate significant differences among samples during the fermentation process (*p* < 0.05).

**Table 2 ijms-25-00684-t002:** Total counts (Log_10_ CFU/g) of microorganisms at days 0, 4, 8 and 12, during fermentation of strawberry tree whole fruits (STWF) and strawberry tree fruit paste (STFP).

Microorganisms	Medium	Days							
		0		4		8		12	
		STWF	STFP	STWF	STFP	STWF	STFP	STWF	STFP
Yeasts	DRBC	2.56 ± 0.99 b	3.58 ± 2.51 b	6.82 ± 3.58 a	7.67 ± 3.13 a	7.07 ± 3.16 a	7.65 ± 4.33 a	7.06 ± 3.33 a	7.65 ± 4.59 a
Moulds	DRBC	3.08 ± 0.72 a	2.68 ± 1.88 a	0 b	0 b	0 b	0 b	0 b	0 b
Aerobic colony count	PCA	3.75 ± 1.23 b	3.66 ± 2.09 a	6.63 ± 3.09 a	2.08 ± 1.48 a	7.85 ± 3.12 a	0 b	6.65 ± 3.51 a	0 b
Lactic acid bacteria	MRS	0 b	0 a	7.19 ± 3.41 a	0 a	7.81 ± 3.87 a	0 a	7.85± 3.83 a	0 a
*Clostridium* spp.	SPS	0 a	0 a	0 a	0 a	0 a	0 a	0 a	0 a
Enterobacteriaceae	VRBGA	0 a	0 a	0 a	0 a	0 a	0 a	0 a	0 a
coli-aerogenes bacteria	VRBA	0 a	0 a	0 a	0 a	0 a	0 a	0 a	0 a
Putative coagulase positive staphylococci	BPA	1.98 ± 0.81 a	2.03 ± 1.18 a	0 b	0 b	0 b	0 b	0 b	0 b

DRBC = Dichloran Rose–Bengal chloramphenicol agar; PCA = plate count agar; MRS = de Man–Rogosa–Sharpe agar; SPS = sulfite polymyxin sulphadiazine agar; VRBGA = violet red bile glucose agar; VRBA = violet red bile agar; BP = Baird Parker agar. Values are the means of three independent measurements ± standard deviation. Letters indicate statistical differences according to the Kruskal–Wallis test followed by Dunn’s multiple comparison post hoc test. a,b—the different letters in lines indicate significant differences among samples during the fermentation process (*p* < 0.05).

**Table 3 ijms-25-00684-t003:** Polyphenols composition of the extracts of strawberry tree whole fruit (STWF) and strawberry tree paste fruit (STFP) not fermented (T0) and fermented (T12), provided by HPLC-DAD analysis.

Polyphenols	STWF		STFP	
	T0	T12	T0	T12
	mg/100 g dw			
Gallic acid der	149.66 ± 10.37 c	117.00 ± 9.30 a	122.45 ± 3.11 a	137.91 ± 4.52 b
Gallic acid	30.03 ± 3.41 b	23.86 ± 7.67 b	16.71 ± 3.70 a	219.87 ± 1.48 c
Ellagic acid der 1	3.12 ± 0.33 a	4.18 ± 0.96 ab	4.51 ± 1.34 b	nd
Ellagic acid der 2	7.04 ± 0.36 b	9.85 ± 2.39 c	10.97 ± 0.61 c	1.94 ± 0.20 a
Ellagic acid	20.61 ± 3.37 a	25.81 ± 2.10 b	17.57 ± 2.61 a	43.67 ± 0.79 c
Quercetin glycoside	2.29 ± 0.42 a	2.83 ± 0.83 ab	3.36 ± 0.78 b	2.81 ± 0.26 ab
Myricetin	0.91 ± 0.11 a	0.81 ± 0.22 a	nd	1.28 ± 0.33 b
Quercetin 3-O-ramnoside	6.06 ± 0.70 a	7.53 ± 0.35 b	7.20 ± 1.32 b	9.71 ± 0.65 c
Ellagic acid der 3	0.79 ± 0.24 b	0.97 ± 0.03 b	0.47 ± 0.17 a	0.87 ± 0.01 b
Quercetin	5.07 ± 1.10 b	5.58 ± 0.95 b	0.84 ± 0.20 a	10.03 ± 2.42 c
Kaempferide	0.53 ± 0.11 b	0.33 ± 0.06 a	nd	0.85 ± 0.18 c
Anthocyanin 1	0.32 ± 0.10 a	1.74 ± 0.11 b	1.42 ± 0.60 b	nd
Cyanidin 3-O-glucoside	4.10 ± 0.40 a	13.31 ± 4.68 ab	19.15 ± 17.71 b	1.60 ± 0.26 a
Anthocyanin 1	0.41 ± 0.15 b	nd	0.26 ± 0.05 a	nd
Anthocyanin 2	1.24 ± 0.30 a	3.94 ± 0.01 b	4.61 ± 0.97 b	1.46 ± 0.56 a
Anthocyanin 3	1.05 ± 0.09 c	0.43 ± 0.08 a	0.46 ± 0.09 a	0.63 ± 0.11 b
Total	233.23 ± 3.10 b	218.17 ± 9.89 a	209.96 ± 19.54 a	432.63 ± 0.67 c

Data are expressed as the mean ± SD of three independent experiments; dw = dried weight; nd = not detectable. Different lowercase letters in the same row indicate statistical differences (*p* < 0.05) in the LSD test.

**Table 4 ijms-25-00684-t004:** Enzyme activities in strawberry tree whole fruit (STWF) and strawberry tree fruit paste (STFP), not fermented (T0) and fermented (T12).

	Lipase U/g	Esterase mU/g	Amylase U/g	Protease U/g	Cellulase U/g	Endo-Xylanase U/g
STWF						
T0	124.05 ± 9.02 a	10.09 ± 0.89 a	19.2 6 ± 1.2 a	40.27 ± 2.80 a	11.30 ± 0.19 a	4.41 ± 0.09 a
T12	166.97 ± 7.64 b	12.31 ± 0.25 b	24.58 ± 2.79 a	53.05 ± 13.23 a	11.89 ± 1.02 a	3.49 ± 0.16 b
STFP						
T0	114.06 ± 4.70 a	5.59 ± 0.89 a	23.34 ± 0.14 a	13.90 ± 2.27 a	12.36 ± 1.58 a	3.48 ± 0.13 a
T12	133.13 ± 12.36 a	9.37 ± 0.22 b	23.68 ± 3.23 a	20.12 ± 2.57 a	9.84 ± 0.95 a	6.06 ± 0.08 b

Letters indicate statistical differences according to the Mann–Whitney test.

**Table 5 ijms-25-00684-t005:** Correlation table of bioactive compounds and enzyme activities (lipase, esterase, amylase, protease, cellulase, endo-xylanase), by a Spearman correlation two-tailed test.

Bioactive Compound	Spearman’s Rho (rs)	Lipase	Esterase	Amylase	Protease	Cellulase	Endo-Xylanase
Total phenol content	Spearman Corr.	0.755240	0.929600 *	−0.083920	0.956220 *	0.237760	0.069930
	*p*-value	0.004510	0.000012	0.795410	0.000001	0.456800	0.829020
Total flavonoid content	Spearman Corr.	0.391610	0.732410 *	−0.524480	0.718040 *	0.132870	0.314690
	*p*-value	0.208060	0.006750	0.080020	0.008540	0.680600	0.319140
Total anthocyanin content	Spearman Corr.	0.237760	0.598610 *	−0.244760	0.605960 *	0.594410 *	−0.531470
	*p*-value	0.456800	0.039740	0.443260	0.036760	0.041520	0.075360
α-Tocopherol	Spearman Corr.	0.584940	0.239860	0.889670 *	0.263160	−0.171630	0.017510
	*p*-value	0.045740	0.452710	0.000107	0.408580	0.593790	0.956920
Lutein	Spearman Corr.	0.447550	0.021130	0.874130	0.038530	−0.447550	0.286710
	*p*-value	0.144590	0.948040	0.000201	0.905370	0.144590	0.366250
Zeaxanthin	Spearman Corr.	0.447550	0.021130	0.874130 *	0.038530	−0.447550	0.286710
	*p*-value	0.144590	0.948040	0.000201	0.905370	0.144590	0.366250
β-carotene	Spearman Corr.	0.711030 *	0.380960	0.742560 *	0.417540	0.073560	−0.101580
	*p*-value	0.009530	0.221790	0.005670	0.176840	0.820280	0.753430

* Correlation is significant at the 0.05 level.

**Table 6 ijms-25-00684-t006:** CAA units of antioxidant activity in extracts from strawberry whole fruit (STWF) and strawberry paste fruit STFP) samples, not fermented (T0) and fermented (T12).

STWF				STFP			
T0		T12		T0		T12	
TPF (µg/g DW)	CAA Unit	TPF (µg/g DW)	CAA Unit	TPF (µg/g DW)	CAA Unit	TPF (µg/g DW)	CAA Unit
240	96	155	96	441	96	441	96
48	93	31	93	88	92	88	92
9.6	77	6.2	86	18	81	18	81
1.92	58	1.24	64	3.5	55	3.5	55
0.38	31	0.25	41	0.70	33	0.70	33

TPF = Total phenolic fraction; CAA = cellular antioxidant activity.

## Data Availability

Relevant data are contained within the article. Additional data are available from the corresponding author.

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
