# Peer review of "Novel Fermentation Strategies of Strawberry Tree Arbutus unedo Fruits to Obtain High Nutritional Value Products"

_ijms, 2024, doi:10.3390/ijms25020684_

Round 1

Reviewer 1 Report

Comments and Suggestions for Authors

Minor revision:

1.       Numerical values must be added to the abstract.

2.       I do not consider it appropriate to describe the effect on the vitamin status of a product in which the vitamin content barely exceeds 1% of the norm in 100 grams of the product in the Conclusion section. I think the authors need to draw the reader’s attention to the high content of flavonoids in the resulting products.

3.       It would also be nice to describe the prospects for using the resulting products in the Conclusion section, based on the results of assessing antioxidant and enzymatic activity.

4.       Line 432 - strange typo

5.       Regarding the resulting cookies, it is not entirely clear whether the tasters liked the taste or not. Was a sample of traditional cookies evaluated? Did they evaluate smell, taste, consistency, traditionalness, color, etc.?

6.       What means ST_B in Figure 6?

Author Response

We thank the reviewer for her/his suggestions.

Minor revision:

  1. Numerical values must be added to the abstract.

We thank the reviewer for pointing this out, numerical values were added to the abstract

  1. I do not consider it appropriate to describe the effect on the vitamin status of a product in which the vitamin content barely exceeds 1% of the norm in 100 grams of the product in the Conclusion section. I think the authors need to draw the reader’s attention to the high content of flavonoids in the resulting products.

We thank the reviewer for this suggestion. The conclusion section was modified

  1. It would also be nice to describe the prospects for using the resulting products in the Conclusion section, based on the results of assessing antioxidant and enzymatic activity.

We agree with this comment and we have accordingly modified the conclusion section

Line 432 - strange typo

 The typo was corrected

  1. Regarding the resulting cookies, it is not entirely clear whether the tasters liked the taste or not. Was a sample of traditional cookies evaluated? Did they evaluate smell, taste, consistency, traditionalness, color, etc.?

 We thank the reviewer for her/his suggestions.

This study was carried out to preliminary assess the possible use of strawberry tree fruits as a source for future applications in food preparations. In our scope, it should offer first evidence to demonstrate a proof of concept for future investigations and we tried to see if the addition of fermented products can enrich or not the sensory traits of a food product.

  1. What means ST_B in Figure 6?

ST_B: standard biscuits produced following the traditional recipe without the addition of strawberry tree fruit preparations. We added the explanation in Figure 6

Reviewer 2 Report

Comments and Suggestions for Authors

This work mainly investigated fermentation of Arbutus unedo fruits and new ingredients obtained with functional properties. The manuscript was also prepared with scientific style and may be interesting article if its global consumption aspects improved, so this manuscript may be recommended for publication in Foods after major revision. Meanwhile, there are some dominant suggested revisions are as follows:

Tittle:

·       The title of the manuscript does not represent the research like comparison of two different way of fermentation, they are not well explained. More clear, fluent and understandable title must be given.

·       It is also suggested that the word strawberry tree be placed before Arbutus unedo

Keywords should be presented in fewer and more effective numbers. There is no need to general terms such as “phenolics”  “flavonoids”  “anthocyanins”  “isoprenoids” and “vitamin”.

Introduction:

The introduction is not covering the whole aspects of topic; it should have sufficient background information to understand and evaluate the study, also, requires to indicate the gap, raising a research question, or challenging prior work in this area and reviewing the pertinent literature to orient the reader.

The novelty of this study should be stated in the text clearly.

Materials and Methods

Additional sub-title "microbial strain" should add to materials and methods section. In this section, it was not clear that from which source the strain was isolated or prepared while this content is given at the beginning of the result and discussion section.

Line 126: In this study, the yeast strain Saccharomyces cerevisiae LI 180-7, previously isolated from fermented black table olives.

Results

The results are correctly defined; however some of them were repeated so that they can come in the introduction or materials and methods.

Table 1 should be redesigned. In this table, the days of fermentation should be modified. Also, the data of two types of fermentation should be side by side in the same table, not as above and below. Its need to present as below:

Time of fermentation (Day)

STWF

STFP

pH

Brix) %)

0

3.61

10.5

4

3.45

8

8

3.23

6.8

12

3.14

5.2

Consistency is not in expressing statistical differences in tables or figures. It should be add the statistic letters in all the some curves. In presenting figures, the statistical difference was not showed.

Others

A review in using abbreviations in the whole manuscript is necessary. For example S. cerevisiae was showed in lines 100, 138 however Saccharomyces cerevisiae LI 180-187, used in lines 126, 433 etc.

Materials and methods section was presented after Results and discussion section and need to be modified. It is necessary to follow the article writing guidelines recommended by the journal.

Author Response

Tittle:

  • The title of the manuscript does not represent the research like comparison of two different way of fermentation, they are not well explained. More clear, fluent and understandable title must be given.
  • It is also suggested that the word strawberry tree be placed before Arbutus unedo

      We thank the reviewer and we have modified the title according to the suggestion

Keywords should be presented in fewer and more effective numbers. There is no need to general terms such as “phenolics”  “flavonoids”  “anthocyanins”  “isoprenoids” and “vitamin”.

The indicated keywords were erased

Introduction:

The introduction is not covering the whole aspects of topic; it should have sufficient background information to understand and evaluate the study, also, requires to indicate the gap, raising a research question, or challenging prior work in this area and reviewing the pertinent literature to orient the reader.

The novelty of this study should be stated in the text clearly.

 The introduction section was revised as suggested.

Materials and Methods

Additional sub-title "microbial strain" should add to materials and methods section. In this section, it was not clear that from which source the strain was isolated or prepared while this content is given at the beginning of the result and discussion section.

We agree with this and have added the “microbial strain” sub-title

Line 126: In this study, the yeast strain Saccharomyces cerevisiae LI 180-7, previously isolated from fermented black table olives.

Results

The results are correctly defined; however some of them were repeated so that they can come in the introduction or materials and methods.

Table 1 should be redesigned. In this table, the days of fermentation should be modified. Also, the data of two types of fermentation should be side by side in the same table, not as above and below. Its need to present as below:

Time of fermentation (Day)

STWF

STFP

pH

Brix) %)

0

3.61

10.5

4

3.45

8

8

3.23

6.8

12

3.14

5.2

Consistency is not in expressing statistical differences in tables or figures. It should be add the statistic letters in all the some curves. In presenting figures, the statistical difference was not showed.

 Table 1 was modified as suggested (please see the revised manuscript version)

Others

A review in using abbreviations in the whole manuscript is necessary. For example S. cerevisiae was showed in lines 100, 138 however Saccharomyces cerevisiae LI 180-187, used in lines 126, 433 etc.

The text was modified as pointed out by the reviewer

Materials and methods section was presented after Results and discussion section and need to be modified. It is necessary to follow the article writing guidelines recommended by the journal.

Material and Methods section is located after the discussion section, as required by the Journal template file

Reviewer 3 Report

Comments and Suggestions for Authors

The authors investigated two strategies for fermentation of strawberry three fruit driven by a selected yeast strain.

The work is well done and described, but there are some weak points. The final aim is not well defined. The comparison of two different preparation for their properties and activities miss the investigation about the repeatability of the batch production and of the fermentation process as well. They also missed to assess the properties keeping of the strawberry three fruit preparation after the production of food prototype, beyond the sensory analysis. I think that the work should be improved before being published.

I have also made some suggestions and changes along the text by Adobe Instrument Comment, please see the attached file.

Author Response

The authors investigated two strategies for fermentation of strawberry three fruit driven by a selected yeast strain.

The work is well done and described, but there are some weak points. The final aim is not well defined. The comparison of two different preparation for their properties and activities miss the investigation about the repeatability of the batch production and of the fermentation process as well. They also missed to assess the properties keeping of the strawberry three fruit preparation after the production of food prototype, beyond the sensory analysis. I think that the work should be improved before being published.

Response:

We thank the reviewer for her/his suggestions.

This study was carried out to preliminary assess the possible use of strawberry tree fruits as a source for future applications in food preparations. In our scope, it should offer first evidences to demonstrate a proof of concept for future investigations. We agree with the productions were made for a single experiment, but at the present time, we would like to individuate the best procedure to ferment these products and then, the repeatability of the process and possible scaling-up is out of the scope of this study.

I have also made some suggestions and changes along the text by Adobe Instrument Comment, please see the attached file.

We really thank the reviewer for her/his suggestions.

The Tables have been modified as suggested. In particular, Table 1 was modified and the statistical analysis was introduced in Tables 1 and 2. The selective medium used for the enumeration of yeasts and moulds, as well as the explanation of all the acronyms of the media was added in Table 2.

Table 3 was modified as requested.

Throughout the text typo errors were corrected as suggested

The letters in Figures 2 and 3 were added.

Figure 6 was improved.